# The Salutogenic Management of Pedagogic Frailty: A Case of Educational Theory Development Using Concept Mapping

**Ian M. Kinchin** 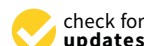

Department of Higher Education, University of Surrey, Guildford GU2 7XH, UK; i.kinchin@surrey.ac.uk

**Abstract:** This paper explores the development of educational theory (pedagogic frailty) that has emerged through the application of concept maps to understand teachers' conceptions of their roles within the complex higher education environment. Within this conceptual paper, pedagogic frailty is reinterpreted using the lens offered by the concept of salutogenesis to place the model in a more positive frame that can offer greater utility for university managers. This development parallels changes in the consideration of mental health literacy (MHL) across university campuses and avoids misapplication of a deficit model to the professional enhancement of teaching quality. For a detailed explication of this wider perspective of pedagogic health literacy (PHL), the connections with related and supporting concepts need to be explained. These include 'assets', 'wellness' and a 'sense of coherence'. Links between these concepts are introduced here. This reframing of the model has used concept mapping to explore the relationship between two complex ideas—pedagogic frailty and salutogenesis. It emphasizes pedagogic health as a continuum operating between frailty and resilience. Brief implications for academic development are included.

**Keywords:** concept mapping; salutogenesis; pedagogic health; academic development; teaching

## 1. Introduction

The current literature on teaching in university is increasingly populated with references about stress and burnout among academics [1,2]. This should raise concerns about the physical and mental health of colleagues working within this system [3], and about the pedagogic health of the higher education system overall. Numerous stressors can be seen to act within the academy. For example, new academics report dissonance between expectations of their role and actual teaching experiences [4]. In addition, competing agendas within universities seem to be adding to the pressures of work [5], while political changes in the system appear to be at odds with the values that drew many academics into academia in the first place [6]:

> Academics are experiencing a growing sense of disconnection between their desires to develop students into engaged, disciplined and critical citizens and the activities that appear to count in the enterprise university. (p. 526)

This generates strong feelings among academics, such as those described so colourfully by Leitch [7] who talks about feeling as though she is "riding two horses at the same time, being propelled simultaneously in opposing directions" (p. 166). The negative consequences of too much stress within the university workforce have been summarised by Mtsweni [8] in his analysis of responses to stress among university administrators:

> the person may attempt to reduce the amount of information to be dealt with by opting for a simplified belief system which denies the true complexity of the issues involved. Typically

this might entail a move towards polarised problem solving with a simplistic yes/no or right/wrong analysis. This diminished judgement can involve an increased personalisation of issues or a hostile egocentricity. In this case the sufferer can only see their limited viewpoint and begins to feel persecuted, interpreting neutral events as being directed at them. Lack of balance is completed by magnification and minimisation whereby trivial are given undue emphasis whilst key factors are played down or ignored. This unsupportable level of cognition eventually leads to fatigue and a state of under-alertness, characterised by forgetfulness, foggy thinking and disorganisation which may be wrongly attributed to a lack of motivation. (p. 20)

Within the context of teaching in higher education, these stress symptoms can be observed to be exhibited by colleagues and these can impede the innovative development of teaching and encourage the rise of 'play safe' classroom practices [9]. Such stress can lead colleagues to consider innovation in teaching practice in a binary manner as either 'good' or more typically 'bad' without considering the wider implications of change and possible benefits to students. The manifestation of hostile egocentricity referred to by Mtsweni can be observed through everyday comments such as, "It just won't work in our department—the management don't understand that we are a special case!" And finally, small changes to relatively minor procedural issues (e.g., which line-spacing should students use in their essays [10]) are often discussed extensively and with passion whilst the 'elephant in the room' is left for another time. The combined effect of these unproductive tensions and workplace stressors that cause 'foggy thinking and disorganisation' can result in an environment exhibiting pedagogic frailty, where elements of the teaching environment seem to be working in opposition to each other so that teachers retreat into a conservative status quo [9] that may be professionally unsatisfying and pedagogically unsound [11]. To address these problems, the model of pedagogic frailty is aligned to key aspects of salutogenesis, to make it more amenable to university managers as a developmental tool. This paper is aimed at those who influence or deliver teaching at university, including teachers, technicians, administrators and managers, as all these roles have an impact on the discourses of learning and on the student experience. Whilst the literature on teaching understandably tends to focus on teachers, it is evident that other roles have an impact on what goes on and how it is reported.

## 2. Pedagogic Frailty and Salutogenesis

The model of pedagogic frailty arose from a fortuitous confluence of personal and professional experiences with a theoretical exploration of university teaching [12]. This drew on three decades of work in teaching and academic development by the author that included several hundred structured teaching observations during which observed teachers often talked about the positive and negative factors influencing their teaching. The author also drew on professional examination of key factors influencing practice in the design of new academic programmes of teacher development [13]. The evolution of the model was also informed by personal encounters with clinical frailty [14] during which the overlap between the literature examining clinical experience and teaching experience became apparent, combined with the theoretical exploration of the visualisation of 'powerful knowledge' [15]. In combination, this gave rise to the conditions in which the model of pedagogic frailty could emerge (Figure 1).

By using the four key dimensions within the model (that have already been explored extensively in the literature [16,17]) to add structure to reflections on teaching practice, so that personal perspectives may be used as a basis for developmental dialogue (such as the example inserted below the model in Figure 1). Although the focus of research on pedagogic frailty considers the university system (i.e., the ways in which the various roles in the institution contribute to teaching), investigations need to start by uncovering the range of perceptions held by individuals within that system. Numerous case studies have revealed the variety of perspectives held by academics across the spectrum of academic

disciplines from the arts to the sciences [17] and the ways in which tensions might develop resulting from conflicting perspectives. The model of pedagogic frailty focuses on four key areas:

- The nature of the discourse on teaching and learning and whether this concentrates on the mechanisms and procedures of teaching (timetabling, assessments, feedback, etc.) or on the underpinning pedagogy (teacher expectations, professional values, student learning approaches, etc.).
- The relationship between the pedagogy and the discipline and whether teaching offers an authentic insight to the discipline in terms of relating theory and practice.
- How the research within the department relates to the teaching in the department, and how these links are exploited in teaching strategies and made explicit in the programme.
- How the teaching is regulated and evaluated and what appreciation there is of the role of the individual academic in the decision-making processes of the institution.

In studies published so far, the model seems to resonate with university academics who readily relate to the idea of frailty and the notion that aspects of the professional environment can create tensions that impede the development of teaching practice [18,19]. However, the negative undertones of the term 'frailty' have been recognised in the clinical literature [20] and may be seen as problematic by university managers when considering the professional development of university academics as it suggests a deficit model. We may attempt to overcome this by adopting an assets-based approach to the consideration of the wider concept of pedagogic health as a continuum linking the extremes of frailty and resilience. The consideration of assets updates the clinical analogy from which pedagogic frailty emerged by offering a parallel to increased consideration of health assets [21,22] within a continuum of health as proposed by Antonovsky in his exploration of salutogenesis [23,24]. Salutogenesis is defined as the study of 'why' and 'how' people stay well [25]. Staying well is related to the ability of individuals to manage tension, that is, how they respond to stressors. The management of tension helps to maintain health. The pathological model is analogous to a deficit model of health, whilst salutogenesis pays more attention to the management of assets that contribute to wellness, and so can be seen to offer links with the ideas of pedagogic health as a continuum between the extremes of pedagogic frailty and pedagogic resilience.

How individuals manage tension and stress in their daily lives and stay well has been referred to as 'salutogenic functioning' [8]. Reframing in this way firstly requires us to discard the dichotomy of diseased/healthy in favour of Antonovsky's health-ease/dis-ease continuum (reframed here as frailty–resilience for the educational context). This is reflected by the application of concept mapping that enables us to visualise nuanced academic perceptions of their 'pedagogic health' [17] in which the diversity of perspectives is valued, and an inappropriate binary good–bad distinction is never made.

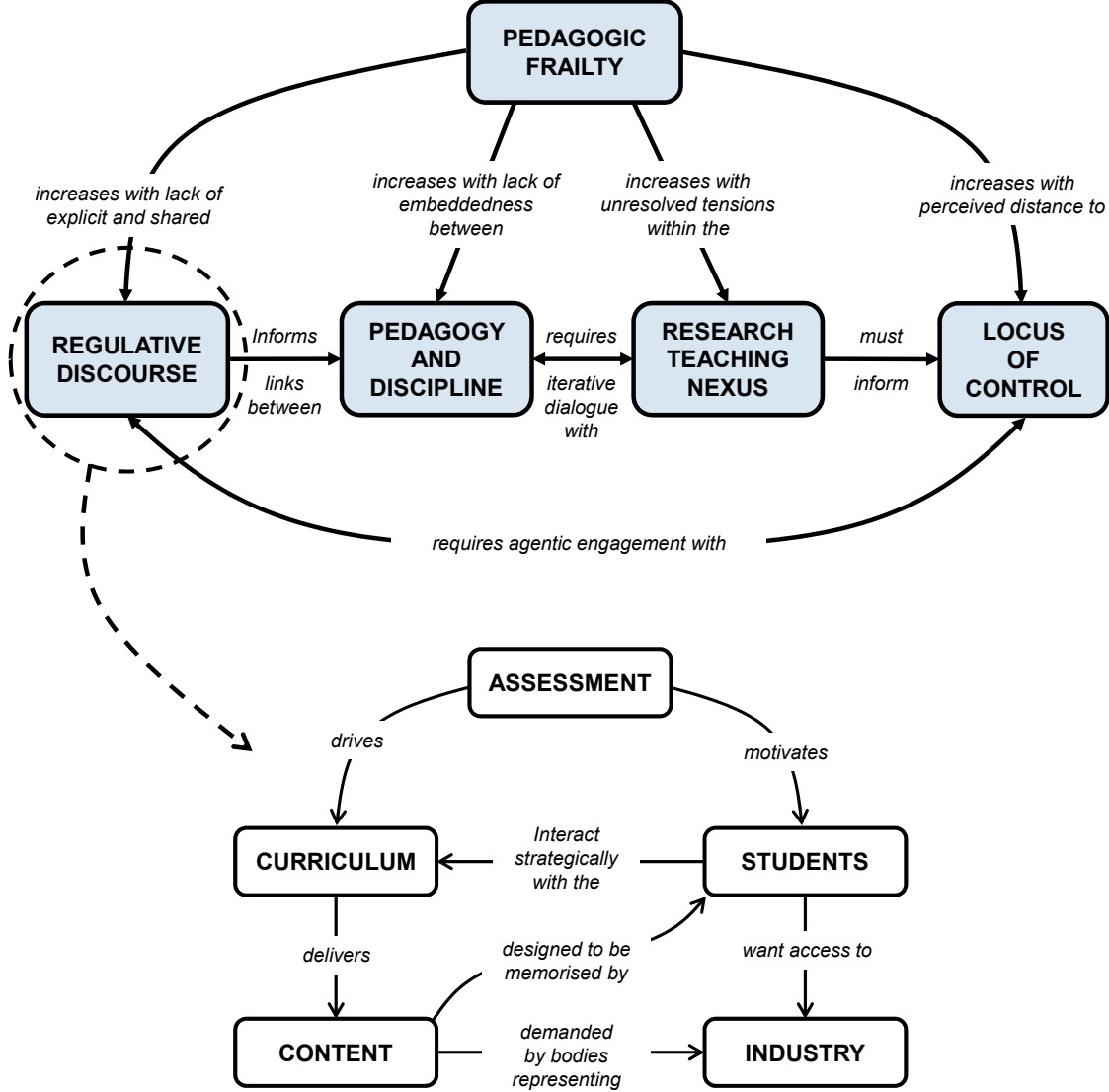

**Figure 1.** The overall pedagogic frailty model (above) with (inset below) one academic's view of the regulative discourse dimension (After Kinchin [15,26]).

Whilst most previous applications of salutogenesis in universities have been concerned with the physical or mental health of individuals working within a university [27], in this paper I have turned this around and am focusing on the health of the system (i.e., the university) where the individuals are working. However, these two perspectives are clearly related to each other and the distinction between a 'healthy academic' and a 'pedagogically healthy' university may be blurred across the numerous interactions between the individual and the institution. The consideration of salutogenesis as a frame for pedagogic health requires a parallel consideration of a number of other associated concepts (particularly assets, wellness and sense of coherence) that need to be part of the network of concepts that will help to generate a robust context to inform practice. Concept mapping offers a tool to allow the visualisation of these ideas and the ways they may be linked (Figure 2).

This visualization of the relationship between salutogenesis and pedagogic frailty represents the author's perspective of the main concepts involved and the relationships between them. The concept map was generated by reducing the problem to include only the main concepts involved and arranged to emphasize the relationships between them. The linking phrases have been constructed to offer the maximum explanatory power in the minimum amount of text in an attempt to produce what has been termed an 'excellent concept map' in the research literature [15]. This provides the reader with a map

to complement the text as a way of reducing cognitive load and making the text more accessible when having to manage a set of unfamiliar terminology.

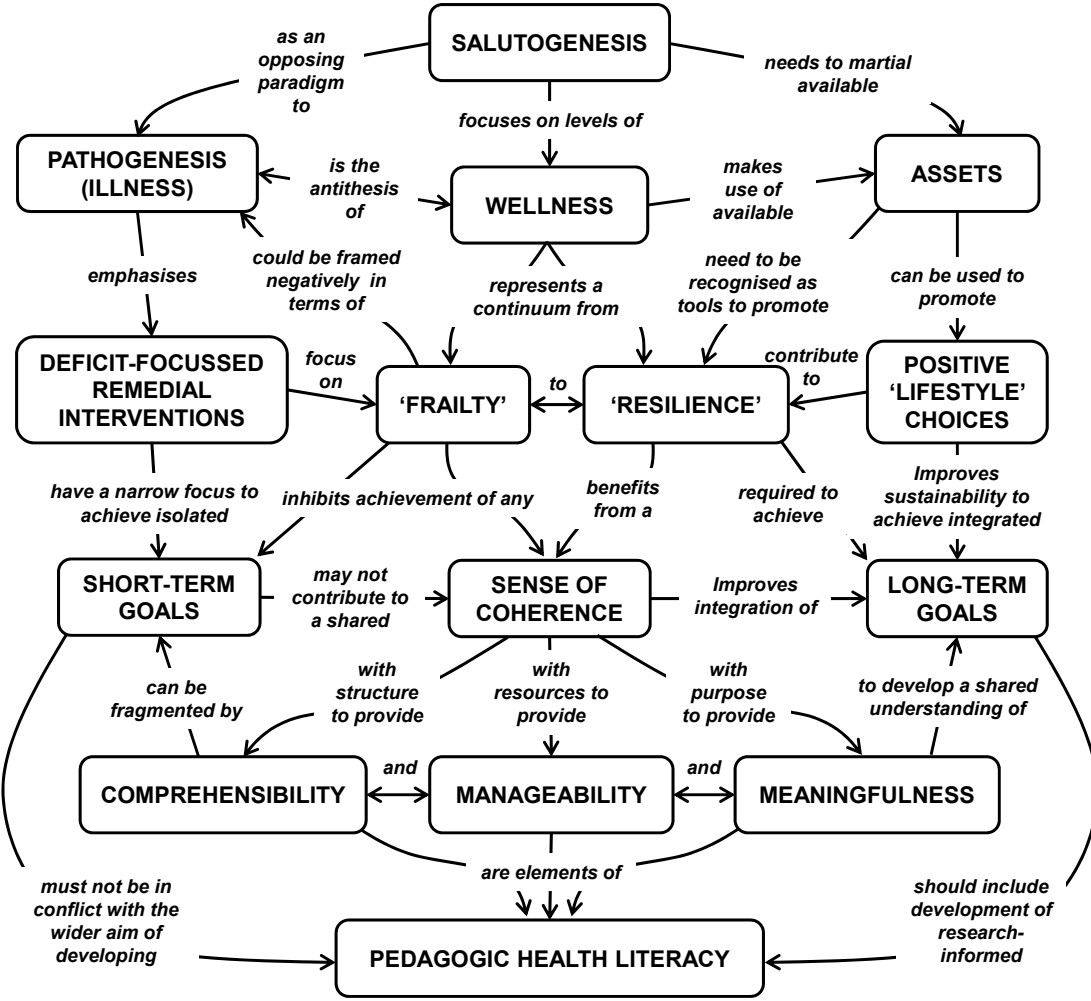

**Figure 2.** A concept map to illustrate the associated concepts that help to relate salutogenesis with pedagogic health literacy.

## 3. Assets

Health assets are starting to be a feature of the healthcare literature that provided the basis for the original frailty analogy. Rotegård et al. [21] define health assets as:

> the repertoire of potentials—internal and external strengths qualities in the individual's possession, both innate and acquired—that mobilise positive health behaviours and optimal health/wellness outcomes. (p. 514)

The assets that people bring to their professional teaching activities can be highlighted and consequently mobilised through reflective dialogue. Some assets may be part of the academic's disciplinary heritage and may be revealed though a process of conceptual exaptation, where familiar disciplinary concepts can be repurposed to support deeper and more personally relevant reflection on teaching (for disciplinary examples, see [28–30], where ideas such as care in nursing, contested concepts in politics and reactions in chemistry are repurposed to provide a language to consider teaching). This reflects the oft-quoted work by Ausubel [31], whose Assimilation Theory of Learning emphasises a constructivist epistemology in which the only place for further learning is provided by what the individual already knows as a basis for the construction of new knowledge. What academics know

best is their disciplinary knowledge and ways of thinking. This echoes the view of frailty management offered by D'Avanzo et al. [32]:

> if we want frailty to be approached as a malleable and preventable condition, a bottom-up approach is needed [and] the tools through which frailty can be managed should come from [participants'] own context and resources. (p. 16)

Rather than providing a rather inert list of assets, a process of 'asset mapping' is suggested in the literature as a process that can help to emphasise the dynamic connections between assets as a way to increase their overall utility [21]. In the case of pedagogic frailty, this asset mapping probably needs to start from the individual perspectives held by academics (as in the case studies illustrated by Kinchin and Winstone [17]) which can then facilitate and structure the essential dialogue between members of an academic community [33,34] to start to map community assets. Distinguishing between individual, community or institutional assets may be helpful in operationalising the pedagogic health model and targeting resources to support the management of a developing sense of coherence [35].

## 4. Wellness

In parallel with the increased focus on health assets within the healthcare literature, there has been an increased focus on the concept of wellness as a way of moving towards health-promoting behaviours. Wellness has been defined by McMahon and Fleury [36] as:

> A purposeful process of individual growth, integration of experience, and meaningful connections with others, reflecting personally valued goals and strength, and resulting in being well and living values. (p. 48)

The idea of 'living values' clearly addresses the comment made by Manathunga in the introduction to this paper, whilst 'meaningful connections with others' is a necessary prerequisite for the mapping of shared assets (such as personal traits or disciplinary skills) mentioned above.

## 5. Sense of Coherence

By linking elements of innovative practice to the frailty model, we are able to support academics in their construction of a greater sense of coherence with regard to the fragmented and contradictory discourses of higher education. Developing a greater sense of coherence within academics of their teaching environments has always been one of the explicit intentions of the application of the pedagogic frailty model [17]. Within the salutogenic paradigm, Antonovsky [23] has defined the sense of coherence (SOC) in terms of its three subcomponents (comprehensibility, manageability and meaningfulness) as:

> a global orientation that expresses the extent to which one has a pervasive, enduring though dynamic feeling of confidence that:
>
> (1) the stimuli deriving from one's internal and external environments in the course of living are structured, predictable and explicable (comprehensibility);
>
> (2) the resources are available to one to meet the demands posed by these stimuli (manageability);
>
> (3) these demands are challenges, worthy of investment and engagement (meaningfulness) (p. 19)

The sense of comprehensibility is supported by consistent, structured information and is confounded by stimuli that are chaotic, random, accidental or inexplicable. Unfortunately, Brookfield [37] reports that some teachers describe their work to be 'bafflingly chaotic'—a situation that augurs badly for the development of a sense of coherence among university academics—presenting a challenge for university managers. A well-developed sense of coherence seems to be related to the ability of academics to cope with stress [38] and is likely to support the development of a positive approach to asset management and wellness.

## 6. Individuals in the System

One important difference between clinical frailty and pedagogic frailty (as previously made explicit in the literature) is that studies of clinical frailty have a focus on the wellbeing of the individual, and consider assessment and treatment of the individual, whereas pedagogic frailty has a broader focus on the system in which that individual operates. This means that any given configuration for an individual may initiate or promote pedagogic frailty in one environment, but promote resilience in another, more receptive, environment. This can be seen in particularly sharp contrast when academics move from one national context to another and find that assets that were valued at home are no longer recognised when they move abroad [39]. However, the structure of individual profiles might predict the potential for frailty, or in other words, certain scripts act as indicators of 'prefrailty'. In an extreme case, a hypothetical, stereotypic academic who is a new arrival at a university might state that "he doesn't care what his colleagues do, he will not adapt any aspect of his teaching to fit current fashions because he has been teaching for twenty years and has established an efficient routine that fits his lifestyle and allows his research to flourish". Such a person might be expected to have difficulties integrating into a new environment that might exhibit a more progressive attitude to innovative teaching approaches. More subtle issues might be predicted where academics map their perceptions of the dimensions of frailty and produce knowledge structures with morphologies that are undeveloped and do not provide sufficient structure to indicate critical reflection on practice. An additional difference between clinical and pedagogic frailty concerns age. Whereas clinical frailty is more prevalent in older patients, pedagogic frailty occurs at any stage of an academic's career and may be repeated as conditions change or as academics take on new roles [40,41].

Research suggests that frailty is a dynamic process that does not sit comfortably in the disease-centred paradigm that dominates medicine [42], and that there are opportunities along its pathway to transition out of, manage and/or prevent its adverse consequences. Considering clinical frailty, Gwyther et al. [43] write:

> Superficially, there appeared to be a dichotomy in beliefs about frailty management. On one hand, some policy-makers appeared to support a greater medicalisation of frailty, a need for frailty to be recognised as an authentic clinical issue by medical professionals and treated as such. On the other, there were views that frailty should be demedicalised and that frailty management should be conceived of as an adaptation to life stages and be embraced as a societal issue with ownership devolved to a wider societal network. (p. 4)

Again, there are direct analogies to be drawn from the comments above to the concept of pedagogic frailty. Rather than a medicalisation of pedagogic frailty, the modern educational world seeks to adopt greater managerialism and accountability to address any frailty in the system, so it may be 'treated'. This is exemplified by the classic "you said, we did" type of management response to student voice. The devolution of management offers a different strategy [44] that would decentralise ownership of the teaching environment that might facilitate frailty management as 'adaptation to professional life stages'.

## 7. Benefits of a Salutogenic Gaze towards Pedagogic Health

By adopting Antonovsky's salutogenic gaze [23,24] to reframe the recent literature on pedagogic frailty [16,17], we might consider the issues that act on teaching in terms of the broader concept of 'pedagogic health'. This requires a modification of the original model of pedagogic frailty (Figure 1) to emphasize the dynamic continuum between frailty and resilience (Figure 3). Introducing the concept of 'pedagogic health' and modifying the linking phrases within the model provides a subtle yet important development for a number of reasons, as it:

- Adopts a more affirmative language (pedagogic health literacy) that may be more appealing to senior managers, having a more positive subtext than frailty.

As an analogy, the increased recognition of mental health issues among both university staff and students has moved from a pathological model (dealing with problems after they have arisen) towards one advocating greater awareness of mental health literacy for all. One of the problems of dealing with student wellbeing within the current Higher Education environment is that 'students approach services when their mental wellbeing is already affecting their ability to cope' [45]. Rather than wait for problems to surface, it may be better to increase the mental health literacy (MHL) of everyone on campus as students with problems also have the potential to affect others including roommates, classmates and staff [3,46–48]. It is, therefore, an issue that affects us all, whatever our own state of mental health. Likewise, before waiting for academics to experience difficulties through frailty within their teaching, moving to the proactive promotion of greater pedagogic health literacy (PHL) across the campus is likely to have a more positive outcome for the institutional community.

• Avoids a potential misuse of the model through adoption of a simplistic harmful binary, the use of which to 'classify' staff would in itself be an indicator of prefrailty.

Within the managerial culture of the neoliberal university, there is pressure to find simplistic, instrumental measures that can be adopted for use as performance indicators [49]. The emerging body of work on pedagogic frailty has demonstrated an underpinning complexity to the teaching environment that cannot be adequately represented by a simple metric. This prevents the concept of pedagogic frailty (or pedagogic health) to be subverted for political means and to prevent the disconnections between expectations and practice described by Manathunga et al. [6].

• Indicates a continuum where no system is likely to exhibit 'total health' and so creates no arbitrary endpoint to prematurely terminate professional development.

The case studies of academics explored by Kinchin and Winstone [17] concentrate on academics who were already recognised as successful teachers. Therefore, each of them has the potential to contribute to pedagogic resilience within their institution. However, I note again here that individual success is not necessarily an indicator of resilience (rather than frailty) across the system, and that even the most successful teaching teams do not exhibit 'total health' (i.e., there is always something new to learn or a new skill to acquire). This depends on developing healthy, positive links between the individuals within a system (e.g., department) for that system to function well.

The learning and development of academics within this perspective do not have a predictable, linear trajectory with an easily defined or predicted endpoint. Rather, '[learning] is an entangled, nonlinear, iterative and recursive process, in which [academics] travel in irregular ways through the various landscapes of their experience (university, family, work, social life) and bring those landscapes into relation with each other' [50]. As such, it resembles the rhizomatic view of learning where knowledge is susceptible to constant modification as it responds to individual or social factors [51].

• The points listed above together help to make utilization of the model more 'management-friendly' and from which management activities are not removed.

It is assumed that senior managers may be reluctant to investigate frailty within the systems over which they preside, and of which they are an active part. The pathological model might be seen as a poisoned chalice. Therefore, by looking at pedagogic health, we have a perspective from which we hope senior managers would not feel the need to exclude themselves—something that would invalidate the whole enterprise.

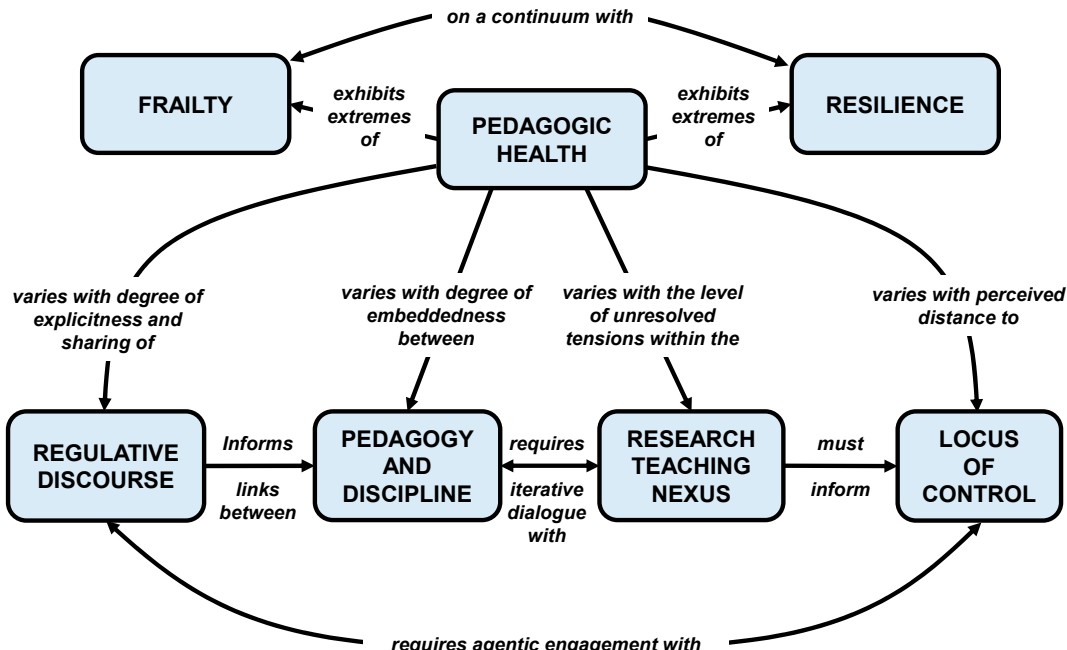

**Figure 3.** A revised model of pedagogic health to indicate the salutogenic continuum between extremes of frailty and resilience.

## 8. Conclusions

The development of a productive teaching environment is key to the success of a modern university. However, in the contemporary economic and political climate of higher education, there are many conflicting and competing discourses that need to be accommodated. Colleagues occupying different roles within the university structure (teacher, administrator, manager) will each have a different perspective on the university—what is important and what should be the institutional focus for the coming years. Inevitably, the development of teaching is a compromise between what we dream of being able to do and what is possible within the current constraints. In order for a university to move forward with a sense of coherence of purpose, the stakeholders within the institution need to have a shared vision of the key elements that make up the teaching environment. When this is achieved, we can talk about the pedagogic health of the institution in positive terms. Where this is not achieved, we might view the pedagogic health in less positive terms—tending towards frailty.

The adoption of appropriate language is an important issue when trying to initiate buy-in from academics (and their managers) to support interventions to enhance teaching quality. Use of the term 'frailty' in this context may invoke fatalistic connotations that were not the intention when originally applying the clinical analogy [26]. Buta et al. [52] have explored the language used in the literature surrounding clinical frailty and have identified some more positive expressions that can be used to better reflect the underlying philosophy of this work (for example, characterizing frailty as 'an opportunity for self-awareness and reflection') that sit better with the original pedagogic frailty model. This can be enhanced using a salutogenic gaze. In order for a salutogenic perspective on PHL to be operationalised within an institution, the network of understanding (Figure 2) needs to be related to productive chains of practice (*sensu* Kinchin and Cabot [53]), that will need to be context-specific. To this end, a modified model of pedagogic frailty is presented (Figure 3). In terms of practical application of this model, the diversity of perspectives revealed by concept mapping is important to help avoid a simplistic binary classification (frail–resilient) as that would preclude the development of individualised routes towards coping and adjustment [32]. Uncovering personal perspectives on the teaching environment [17] provides a shared lexicon to open up the potential for more collaborative discussions, helping to promote the development of a shared perspective underpinned by a set of

common values [54]. This is likely to promote more productive dialogue about teaching, and the promotion of an environment that supports greater pedagogic health.

Within the author's university, moves towards a more salutogenic perspective of pedagogic health currently involve initiatives such as supporting a gradual transition from a 'responding-to-student-voice' mode of operation towards a greater degree of 'student–staff partnership' [55], as indicated in Figure 2 as a 'life-style choice' for the institution. Such a move is seen to have 'the potential to remedy neoliberal university models and performance self-regulation by offering a counternarrative to these dominant trends that imagine a different model of learning between students and staff' [56]. Analysis of the practicalities of developing a salutogenic gaze towards PHL across a university is not anticipated to be straightforward within institutions that have developed particular cultures and ways of doing things over many years. It is particularly important that the senior managers in an institution demonstrate a commitment to the counternarrative to ensure that the values that are espoused by the teachers working in partnership with the students are reflected in the language used by and actions demonstrated by the management [57].

The clinical literature continues to provide the basis for analogy in describing the variation in the way that frailty is experienced by individuals as they struggle to maintain their daily routines in the context of complicated transitions that create uncertainty [58]. Work on pedagogic frailty suggests a need to raise awareness among professionals (academics and university managers) of the malleability and preventability of frailty and the benefits of having an informed 'navigator' [43] or 'interpreter' [59]—which, in the case of pedagogic frailty, would be an academic developer [60] or learning developer [61]—to assist in steering a route through appropriate interventions. This provides a modified perspective in the role of the academic community as a whole, through a more distributed ownership of pedagogic health [44]. It also requires academic developers to be in a position to support the process of conceptual exaptation of disciplinary concepts for the enhancement of teaching. This may be achieved most effectively when academic development becomes a distributed activity [62], and those involved are actively engaged in research with their disciplinary peers [63]. This still requires effort on the part of all stakeholders to engage in professional development to ensure a positive direction of travel along the frailty—resilience continuum. As explained by D'Avanzo et al. [32], 'The resilience of people who succeed is achieved through continuous active development on numerous fronts' (p. 15).

In summary, it makes no sense to consider interrelated aspects of university teaching in isolation. Educational theory and analysis of practice needs to be considered as a connected whole. The salutogenic model of pedagogic health provides a lens to help achieve this goal, and concept mapping provides a tool that emphasises connectivity. Jonas [64] has referred to salutogenesis as an 'anchoring principle' to unify all dimensions of a healthcare system. I suggest here that salutogenesis offers great potential as a concept that can be repurposed to add clarity and a sense of coherence to the teaching endeavour. This added coherence is particularly important in the current context of political and economic uncertainty about the future of higher education and the need to care for our students [65]. To reinterpret comments by Becker et al. [66] by placing them into the discourse of teaching, 'A salutogenic gaze works prospectively by considering how to create, enhance, and improve resilience and provides a framework for researchers, and practitioners to help individuals, and organizations to move towards optimal pedagogic health' (p. 25). This potential seems worth exploring, and the application of concept mapping offers a nuanced methodology to do this.

**Funding:** This research received no external funding.

**Conflicts of Interest:** The author declares no conflict of interest.

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
