# Peer review of "The Salutogenic Management of Pedagogic Frailty: A Case of Educational Theory Development Using Concept Mapping"

_education, doi:10.3390/educsci9020157_

Round 1

Reviewer 1 Report

The concept of pedagogic frailty needs clarifying and perhaps also defining at the outset (is the line starting on page 2 line 59 an attempt at a definition?). I’m familiar with key papers on this topic but other readers may not be.

There’s a heavy reliance on quotes and the one on page 4 has no page numbers. consider reducing quotes.

The audience for the paper seems to waver between administrators, making things more palatable for senior managers, academics and academic developers, it’s unclear who the paper is aimed at, and especially what they might do as a result which should be clear from the conclusion.

The way that the figures were created is unclear e.g. figure 1 is described on page 2 lines 68 and 69 as simply ‘emerging’. It should be clarified as to why the model was developed and for whom (and by whom? Ie what role, under what conditions). The same goes for figure 2. How was it elicited? The wording of the paper makes it sound like concept mapping is a methodology, but it’s unclear how it was employed.

 If the basis of the paper is that stress impedes the development of teaching practice (page 2 lines 49-50), some evidence should be provided for this.

Describing the university as a system (page 3 line 102), is fuzzy at this point. What aspect of the system is the focus – staff? Students? Buildings? Reputation?. This becomes a bit clearer in section 6 (page 6) but the system idea could be introduced and clarified earlier. In section 6 the source of material in lines 184-190 is unclear. The quote on lines 200-205 seems less relevant.

 Page 8 line 249 ‘teams does’ replace with either teams do or team does

Author Response

Review 1

Comments and Suggestions for Authors

The concept of pedagogic frailty needs clarifying and perhaps also defining at the outset (is the line starting on page 2 line 59 an attempt at a definition?). I’m familiar with key papers on this topic but other readers may not be.

Additional clarification has been inserted here to address this. However, I am also keen not to simply repeat material that has been published elsewhere and is referenced here. I hope I have now achieved a better balance. Additional explanation has been added to pp. 2-3.

There’s a heavy reliance on quotes and the one on page 4 has no page numbers. consider reducing quotes.

There are a number of quotes used in the text, however these have been carefully selected to emphasise the ways in which the key ideas that underpin clinical frailty can be repurposed to help in the articulation of pedagogic frailty. The missing page number for the quote on page 4 has been added.

The audience for the paper seems to waver between administrators, making things more palatable for senior managers, academics and academic developers, it’s unclear who the paper is aimed at, and especially what they might do as a result which should be clear from the conclusion.

Comments have been inserted to add clarity here regarding the complementary roles of various stakeholders for the development of a teaching environment.

The way that the figures were created is unclear e.g. figure 1 is described on page 2 lines 68 and 69 as simply ‘emerging’. It should be clarified as to why the model was developed and for whom (and by whom? Ie what role, under what conditions). The same goes for figure 2. How was it elicited? The wording of the paper makes it sound like concept mapping is a methodology, but it’s unclear how it was employed.

Comment is inserted to add clarity here to indicate that this ‘emergence’ was actually based on several years of work and represents a synthesis of earlier research. The rationale for the nature of figure 2 is added on page 4.

 If the basis of the paper is that stress impedes the development of teaching practice (page 2 lines 49-50), some evidence should be provided for this.

The use of the literature is clarified here by reordering references 9 and 10, and bringing elements discussed in 10 to the fore.

Describing the university as a system (page 3 line 102), is fuzzy at this point. What aspect of the system is the focus – staff? Students? Buildings? Reputation?. This becomes a bit clearer in section 6 (page 6) but the system idea could be introduced and clarified earlier. In section 6 the source of material in lines 184-190 is unclear. The quote on lines 200-205 seems less relevant.

Comment is inserted to clarify what is meant by the ‘system’, introducing the idea on page 2.

 Page 8 line 249 ‘teams does’ replace with either teams do or team does

This typo is corrected to ‘teams do’.

I thank the reviewer for the thoughtful and constructive comments. Additions to the manuscript are given in red.

Reviewer 2 Report

This article has consistency with what they propose to do and what they describe having done. The part of the conceptual structure to illustrate the associated concepts (Figures 3) is very well defined. However, the authors should explain in section 3, the theory of Ausebel related to the learning concerning their research.

Author Response

Review 2

This article has consistency with what they propose to do and what they describe having done. The part of the conceptual structure to illustrate the associated concepts (Figures 3) is very well defined. However, the authors should explain in section 3, the theory of Ausebel related to the learning concerning their research.

The essence of Ausubel’s theory has been added here on pages 5 and 6.